# How May Obesity-Induced Oxidative Stress Affect the Outcome of COVID-19 Vaccines? Lesson Learned from the Infection

**Claudia Pivonello [1],\*** , **Mariarosaria Negri [1]** , **Rosario Pivonello [1,2]** and **Annamaria Colao [1,2]**

[1] Dipartimento di Medicina Clinica e Chirurgia, Federico II University of Naples, 80131 Naples, Italy; negrimariarosaria@yahoo.it (M.N.); rosario.pivonello@unina.it (R.P.); colao@unina.it (A.C.)

[2] UNESCO Chair for Health Education and Sustainable Development, Federico II University of Naples, 80131 Naples, Italy

\* Correspondence: claudia.pivonello@unina.it; Tel.: +39-0817464737

The coronavirus disease 2019 (COVID-19) outbreak, caused by the severe acute respiratory syndrome coronavirus 2 (SARS-CoV-2), has induced a global emergency [1]. The rapid dissemination of the virus, the severity of the disease, and the increasing number of deaths have necessitated the reinforcing of public health measures by local health authorities worldwide. Particularly in Italy, the limited capacity of hospitals was among the most challenging issues facing authorities, placing an excessive burden on the Italian healthcare system [2]. Nevertheless, surprisingly, in less than 12 months, researchers sequenced SARS-CoV-2′s genome and established several randomized controlled trials to test numerous vaccines, based on different technologies [3,4], some of which are currently under development while others have been now proven to be effective, with different percentages of efficacy, in protecting against illness. The aim of COVID-19 vaccines is to induce a humoral and cellular immune response preventing the morbidity and mortality induced by COVID-19 [5], and therefore avoiding the collapse of hospitals and healthcare systems. Humoral immune response to SARS-CoV-2 is mediated by immunoglobulins (or antibodies), produced by B lymphocytes which mature into short-lived plasma cells, against different viral surface proteins [6], while cellular immune response is mediated by CD4+ T (T helper) and CD8+ T (T suppressor) lymphocytes and B lymphocytes [3].

As already observed during the influenza A H1N1 pandemic in 2009 [7], obesity has also been recognized as an independent risk factor of severe disease, hospitalization, and death during the COVID-19 pandemic [8–12]. Obese patients are characterized by hyperglycaemia, hyperlipidaemia, vitamin and mineral deficiencies and high levels of bioactive adipokines, such as leptin, which are all factors contributing to oxidative stress (OS), a condition of imbalance favouring the reactive oxygenated species (ROS) including superoxide anion ($O_2^{\bullet-}$), hydrogen peroxide ($H_2O_2$) and hydroxyl radicals ($OH^\bullet$), compared to antioxidants [13]. In obese subjects, intracellular and plasma glucose and free fatty acids overload promotes the generation of superoxide radicals by activating the redox-sensitive transcription factors NADPH oxidases (NOX) [13]. ROS can modify the activity of several transcription factors, including the nuclear factor kB (NF-kB), involved in the activation of pro-inflammatory cytokines and chemokines as well as in inflammasome regulation, by affecting the host's immune response [14]. Concomitantly, the reduced sources of antioxidants, including superoxide dismutase, glutathione peroxidase, catalase, vitamin A, E, C and carotenoids in obese patients promotes a vulnerability to oxidative damage and consequently increases susceptibility to infections [14].

Moreover, obesity is associated with a state of metaflammation—chronic low-grade inflammation—a condition that, among other factors, contributes to inducing systemic OS. Indeed, elevated urinary levels of F2-isoprostanes, prostaglandin-like compounds which are products of the peroxidation of polyunsaturated fatty acids involved in pro-inflammatory mechanisms and biomarkers of in vivo OS, have been found in obese patients [15]. An additional feature of metaflammation is the increased infiltration of im-

mune cells, including macrophages, mast cells and natural killer, into adipose tissue, contributing to the worsening of the inflammatory milieu—already compromised by the dysregulated production of adipokines [16,17]. Indeed, obesity is characterized by reduced levels of adiponectin, an anti-inflammatory adipokine, and increased levels of leptin, an adipokine with mainly pro-inflammatory properties. Characteristic hyperlipidaemia observed in obese patients stimulates monocytes and macrophages and induces the production of pro-inflammatory cytokines such as tumour necrosis factor (TNF)-$\alpha$ and interleukin (IL)-6 [16,18,19]. Moreover, in obese COVID-19 patients, the already pre-existing effects prompted by the lipid peroxidation-dependent OS could be further aggravated by SARS-CoV-2 infection [20], affecting the immune control system in response to infection and potentially increasing the severity of the lung disease and contributing to multiorgan failure. Indeed, as already demonstrated in several virus infections, during SARS-CoV-2 infection, the overproduction of $H_2O_2$ released by infected neutrophils [21] could induce the activation of NF-kB in macrophages and T lymphocytes, promoting the secretion of pro-inflammatory cytokines, including TNF-$\alpha$ and IL-6 [22]. Particularly, OS distresses the cell structure of T lymphocytes by weakening the plasma membrane and the T lymphocytes' function and number, with a loss of reactivity and viability [23,24] and through an apoptotic mechanism [25], respectively. It is therefore reasonable to assume that the lymphocytopenia, observed in obese patients in severe COVID-19 settings [26], is ascribable to the pre-existing OS causing impaired phagocytosis, less efficient CD4+ T lymphocytes, as well as the reduced number and activity of CD8+ T lymphocytes and B lymphocytes—the latter producing a low titre of antibodies and decreased antibody neutralization capability [14], probably contributing to the coagulopathies and cardiovascular co-morbidities observed in severe disease [5].

These conditions dampen and delay the immune response to viral infections, raising the question of whether the COVID-19 vaccine can be as effective in obese patients as in lean patients.

Previous studies conducted to investigate the response to several vaccinations (hepatitis A and B, tetanus and influenza A vaccines) in obese patients demonstrated that high body mass index (BMI) is associated with a reduced or total lack of antibody response to the hepatitis B vaccine, reduced seroconversion following hepatitis A vaccine, elevated IL-6 levels and low antibody response to tetanus vaccine, as well as low CD4+ and CD8+ T lymphocytes function, low antibody response and reduced seroconversion following influenza A vaccine (reviewed in [27]). To date, only one non-peer-reviewed Italian study investigated the humoral response in a cohort of 248 healthcare workers, undergoing an mRNA vaccine (BNT162b2, Pfizer), 7 days after the second dose [28]. The results of the study demonstrated that the antibody titre was significantly higher in young and female participants compared to the male and older population. Moreover, the humoral response was significantly more efficient in subjects with lower and normal weight compared to overweight and obese subjects. In conclusion, the consequence of OS on immunological functions and the evidence of weakened virus vaccine effectiveness in obese patients raise concerns about COVID-19 vaccine responsiveness in this population. Further clinical studies are mandatory to investigate a potential different and long-lasting humoral and cellular response following the encapsulated mRNA or viral vector anti COVID-19 vaccines in obese subjects.

**Author Contributions:** Conceptualization, C.P.; literature search C.P. and M.N.; writing—original draft preparation, C.P. and M.N.; writing—review and editing, R.P. and A.C.; supervision, A.C. All authors have read and agreed to the published version of the manuscript.

**Funding:** This research received no external funding.

**Institutional Review Board Statement:** Not applicable.

**Informed Consent Statement:** Not applicable.

**Conflicts of Interest:** The authors have no conflict of interest to declare.

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
