# Peer review of "How May Obesity-Induced Oxidative Stress Affect the Outcome of COVID-19 Vaccines? Lesson Learned from the Infection"

_stresses, doi:10.3390/stresses1020010_

Round 1

Reviewer 1 Report

This piece is very interesting. I believe that, if there are data in the literature, the authors could do a comparison between the different vaccines.

Author Response

We thank the reviewer for his/her interesting comments. Unfortunately, just one paper (cited in the perspective) reporting the cellular and humoral immune response in patients with obesity is present in literature, therefore a comparison between the different vaccines can not be performed and discussed.

Reviewer 2 Report

Thanks for your perspective submission on how obesity-induced oxidative stress may affect the outcome to COVID-19 vaccines. This is a useful submission as obesity has been known to influence the possibility of having a more severe COVID infection and influences disease outcomes. Your submission sheds light on the plausible mechanisms/oxidative stress.

Below are my comments:

1) There are typos and syntax errors in the submission that require correction.

2) References are not listed. Please provide the references.

3) Certain phrases and sentences need to be rewritten for better clarity.

4) Below references may be useful to include if not included already

  1.  Body Mass Index and Risk for COVID-19–Related Hospitalization, Intensive Care Unit Admission, Invasive Mechanical Ventilation, and Death — United States, March–December 2020- Kompaniyets et al
  2.  Kates J, Dawson L, Tolbert J.Kaiser Family Foundation.The next-phase of vaccine distribution:High-risk medical conditions. https://www.kff.org/policy-watch/the-next-phase-of-vaccine-distribution-high-risk-medical-conditions/
  3. Obesity and mortality of COVID-19. Meta-analysis-Hussain et al
  4. Obesity and impaired metabolic health in patients with COVID-19- Stefan et al

Author Response

We thank the reviewer for his/her remarks. 

We apologize for the lack of reference list. We reported it in the bottom of the file and we also added the references suggested by the reviewer. Unfortunately, the reference "Obesity and mortality of COVID-19. Meta-analysis-Hussain et al" has been retracted by the journal and thus we can not cited it.

Typos and syntax errors have been corrected as reported in R1 file. 

Reviewer 3 Report

No specific comments

Author Response

We thank the reviewer for his/her appreciation of our work.